# Consistent3DGen: Bridging Stochastic Generation and Deterministic Reconstruction for Image-to-3D Diffusion Models

## Abstract

Recent large-scale 3D diffusion models have achieved remarkable success in generating high-quality and detailed 3D objects. However, due to their reliance on randomized initial noise sampling, these models often produce 3D objects that, while visually similar to input images, lack precise consistency with them. We attribute this limitation to the fundamental tension between generative modeling and faithful reconstruction. We argue that the image-to-3D task should be the combination of reconstruction at known views and completion at unknown views. To address this challenge, we propose Consistent3DGen, a training-free framework that ensures consistency for existing 3D diffusion models. Our approach leverages state-of-the-art pixel-aligned point cloud reconstruction algorithms, such as VGGT, to obtain geometrically consistent 3D point clouds from input images. We then introduce a mechanism to map these front-facing point clouds into the VAE latent space of 3D diffusion models, and design a novel algorithm for completing the back part by front-partial denoising guidance. Extensive experiments demonstrate that our method achieves high consistency in the face-forward direction of 3D models, especially in situations where consistency is required, *e.g.*, characters.

## 1 Introduction

The recent release of large-scale 3D datasets, particularly Objaverse (Deitke et al., 2023b;a), has catalyzed significant advances in 3D diffusion models (Lu et al., 2024; Li et al., 2025b). These developments have profound implications for automated 3D asset creation, promising to substantially reduce costs across diverse industries, including gaming, e-commerce, film production, and beyond.

Recent advances in 3D diffusion models have explored diverse latent representations to enhance generation quality and controllability. Early approaches (Zhang & Wonka; Zhang et al., 2024) primarily employed latent sets or VecSets (Zhang et al., 2023) as the foundational representation for VAE latent spaces. Subsequently, a paradigm shift toward sparse voxel has emerged, enabling extremely high spatial resolution and achieving remarkable fidelity in geometric details (Ren et al., 2024b; Xiang et al., 2025; Wu et al., 2025; Li et al., 2025b; Chen et al., 2025). These explicit representations also provide enhanced controllability. Despite these significant advances, a critical limitation persists: the stochastic nature of initial noise sampling leads to substantial variability in the generated 3D objects, particularly when viewed from the input image perspective. This inherent randomness, while beneficial for diversity, fundamentally undermines the user's ability to obtain a precise 3D object that they want. This phenomenon can be characterized as a form of 3D visual hallucination, analogous to hallucination problems in the field of image generation (Lyu et al., 2025) and LLM (Achiam et al., 2023).

We identify this limitation as stemming from a fundamental tension between generative modeling and faithful reconstruction. Generative modeling prioritizes learning rich distributions. Its stochastic nature, which relies on sampling from random initial noise, inherently introduces variability that enhances creative generation. In contrast, faithful reconstruction demands strict adherence to the observed evidence, requiring the model to precisely match the input view. This contradiction presents a significant challenge: how can we harness the powerful generative capabilities of diffusion models while ensuring strict consistency with input observations?

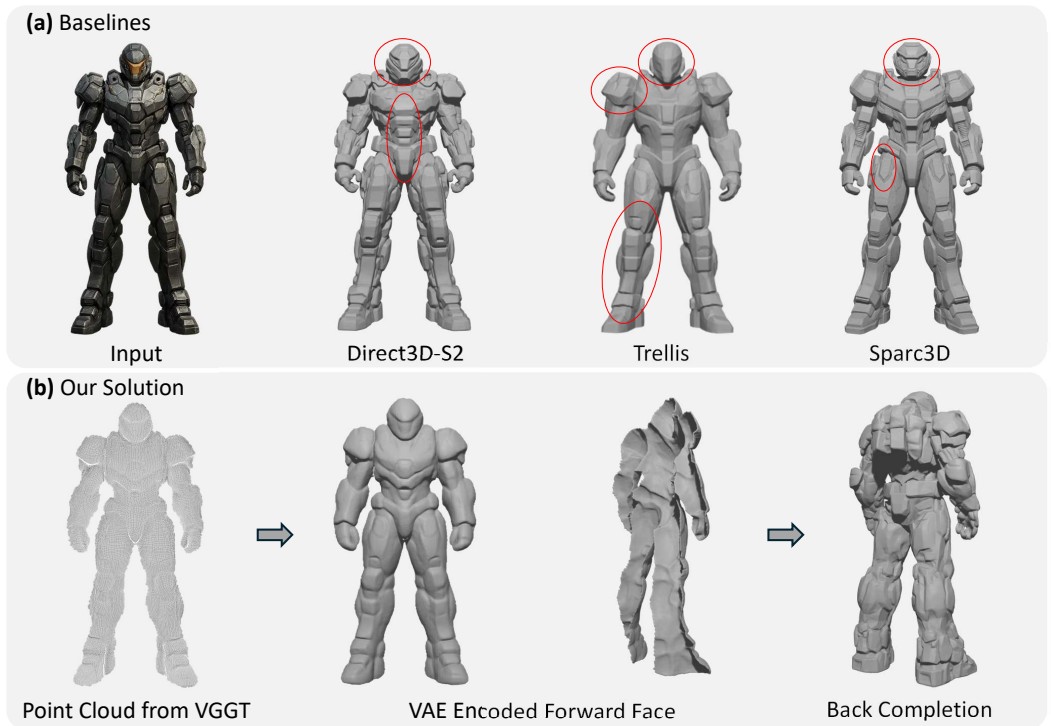

Figure 1: (a) Though baseline methods can produce highly detailed 3D objects from images, there is still a lack of **detail-consistency**. (b) We introduce a pixel-aligned 3R model, VGGT, to produce a consistent foreground surface, while using a diffusion model to complete its back.

To address this fundamental challenge, we propose a training-free framework, that elegantly reconciles the precision of deterministic reconstruction with the creative power of generative diffusion. Our key insight lies in decoupling the reconstruction and generation tasks: we leverage state-of-the-art deterministic reconstruction models to establish geometrically faithful constraints for visible regions, while preserving the diffusion model's generative capabilities for completing unobserved areas. Specifically, we harness recent advances in pixel-aligned reconstruction, such as VGGT (Wang et al., 2025a), which produces highly accurate front-facing 3D surface point clouds. To integrate these geometric priors into the denoise process, we develop a novel point-to-latent mapping algorithm that transforms the front-facing surface point cloud into the latent space of 3D diffusion models. Building upon this foundation, we introduce a front-partial denoising guidance mechanism that strategically steers the diffusion process to converge toward the predetermined partial latent for visible regions, while allowing unconstrained generation for occluded areas. This dual-mode approach ensures that the final 3D output maintains strict consistency with the input view without compromising the model's ability to synthesize plausible completions. Extensive experiments validate the effectiveness of our framework.

To summarize, our contributions are listed as follows:

- We propose Consistent3DGen, a training-free framework that guarantees consistency between input views and generated 3D objects.
- We design an algorithm to map the surface point cloud to partial latent and a front-partial denoising guidance mechanism.
- We propose a method that can complete the back part for a given foreground geometry.

## 2 RELATED WORKS

**3D Diffusion Models.** With the introduction of a large-scale 3D dataset (Deitke et al., 2023b), 3D Generative Models achieve remarkable performance in image-to-3D generation tasks. In these methods, the most prominent direction is diffusion-based models, achieving unbelievable details gener-

ation. 3DShape2VecSet (Zhang et al., 2023) proposes a VAE architecture, enabling the 3D shape compression with latent representation. Based on this latent design, LaGeM (Zhang & Wonka), Clay (Zhang et al., 2024), CraftsMan3D (Li et al., 2025a), *etc*, propose different diffusion model designs from different aspects. Besides, the latent set, some researchers focus on sparse data structures, such as sparse voxels, which only store and compute at the partial voxels. These methods can generate much higher spatial resolution 3D models by removing the cost for empty voxels in the space. XCube (Ren et al., 2024a) is the first work to train VAE and Diffusion based on sparse voxel representation. Trellis (Xiang et al., 2025) designs a representation, Structured LATent (SLAT) architecture, as the data structure for the VAE latent space. To further release the potential of the sparse voxel, Direct3D-S2 (Wu et al., 2025) develops a simplified self-attention mechanism to reduce the computational cost introduced by $\mathcal{O}(n^2)$ complexity in vanilla self-attention, supporting $1024^3$ spatial resolution training. Ultra3D (Chen et al., 2025) also designs a similar part-level self-attention mechanism. Sparc3D (Li et al., 2025b) optimizes the dense-based mesh extraction algorithm for sparse voxel. Different from maximizing the spatial resolution, Hi3DGen (Ye et al., 2025) proposes to utilize the normal maps as the bridge from RGB to 3D, achieving higher 3D accuracy. In this paper, we propose an algorithm that can make the generated mesh aligned with the input image in a training-free manner. Therefore, following the progress of 3D diffusion models, our method can evolve together.

**Other 3D Generative Models.** Besides diffusion models, there are some other kinds of 3D generative models, such as GAN (Schwarz et al., 2020; Jiang et al., 2023; Chan et al., 2021), feed-forward models (Hong et al., 2023; Jiang et al., 2025; Xu et al., 2024), auto-regressive models (Siddiqui et al., 2024; Tang et al.; Zhao et al., 2025), *etc*. In the early era of 3D generative models, there were not large-scale 3D datasets. Therefore, the researchers train 3D-aware GAN models (Goodfellow et al., 2014) using 2D image datasets, like FFHQ (Karras et al., 2019). These models can only generate 3D models for a single class. After the release of a large-scale 3D dataset, Objaverse (Deitke et al., 2023b;a), LRM (Xie et al., 2024) is the first method that trains a single-image-to-3D feed-forward model for arbitrary objects' generation. Following this pipeline, numerous works are making progress. For example, InstantMesh (Xu et al., 2024) implements a mesh feed-forward model from 6 sparse views. MeshFormer (Liu et al., 2024) re-designs the framework based on sparse voxel to improve the spatial resolution. PRM (Ge et al., 2024) introduces the physically-based rendering into this pipeline for photorealistic input and rendering. Nonetheless, these methods struggle in generating detail-abundant meshes compared with diffusion models. More recently, auto-regressive models have received huge attention due to the great success of Large Language Models. Different from previous methods, with an iso-surface extraction algorithm, auto-regressive models aim at real-world application generation. However, limited by the tokenization algorithm for mesh faces, the useful training data is limited and can not generate fine-detail meshes with a large number of faces. Considering these disadvantages, we focus on improving the 3D diffusion models.

**3D Reconstruction Models.** Different traditional 3D reconstruction methods, such as Structure-from-Motion and Neural-based per-scene optimization methods (Mildenhall et al., 2021; Kerbl et al., 2023), recent methods exhibit a clear evolution towards transformer-based and pixel-aligned generic feed-forward models. For each pixel in the input image, the 3D reconstruction models predict the point coordinate in 3D space. A pioneering method, DUSt3R (Wang et al., 2024), introduces a transformer-based paradigm that requires no prior camera calibration, effectively relaxing classical projective constraints and unifying monocular and stereo reconstruction in one framework. Subsequent works improve this method from many aspects, including accuracy (Smith et al., 2025; Dong et al., 2025) and scene scale (Yang et al., 2025; Wang et al., 2025b;a). Among these methods, VGGT (Wang et al., 2025a) achieves the current best performance both on the large-scale scene reconstruction and sparse-view reconstruction, demonstrating a huge potential. The pixel-aligned nature significantly facilitates the accuracy corresponding to the input image. Therefore, we incorporate this nature into 3D generative models for a better alignment to user input.

## 3 METHOD

In this section, we first introduce the necessary components of our training-free inference system (Sec. 3.1). Then, we introduce our whole pipeline (Sec. 3.2).

## 3.1 BACKGROUND

### 3.1.1 SPARSE VOXEL

Unlike conventional dense voxel grids that allocate memory for every cell in the volume, a sparse voxel representation stores only the occupied cells. Formally, a dense grid with spatial resolution $H \times W \times D$ and $C$ feature channels is a tensor $X \in \mathbb{R}^{H \times W \times D \times C}$. In contrast, the sparse form comprises (i) an index tensor $I \in \mathbb{N}^{N_a \times 3}$ containing integer coordinates $(x, y, z)$ of the $N_a$ active voxels, and (ii) a feature tensor $F \in \mathbb{R}^{N_a \times C}$ storing the corresponding features. For example, when $H = W = D = 64$ and $C = 16$, the dense representation is $64 \times 64 \times 64 \times 16$, whereas if only $N_a = 30,000$ voxels are active, the sparse representation stores a $30,000 \times 3$ index tensor and a $30,000 \times 16$ feature tensor. This design substantially reduces memory footprint and, by restricting computation to the active set, avoids unnecessary operations in empty regions; consequently, both storage and compute scale with $N_a$ rather than with the full grid volume $H \times W \times D$.

### 3.1.2 SPARSE-VOXEL-BASED 3D DIFFUSION MODELS

We begin by explaining the sparse-voxel-based 3D diffusion model. We chose Direct3D-S2 (Wu et al., 2025) as our base model, the SoTA publicly available 3D diffusion model. This model uses a flow matching approach with three stages: one dense stage followed by two sparse super-resolution stages. The model works through the following steps:

**Stage 1: Dense Processing** First, we sample a random Gaussian noise:

$$x_t^{\text{dense}} \in \mathbb{R}^{16 \times 16 \times 16 \times 16} \sim \mathcal{N}(0, 1). \tag{1}$$

Given an input image $I \in \mathbb{R}^{H \times W \times 3}$, a flow-matching DiT model $\epsilon^{\text{dense}}(\cdot)$ denoises $x_t^{\text{dense}}$ iteratively:

$$x_{t-1}^{\text{dense}} = x_t^{\text{dense}} - \epsilon^{\text{dense}}(x_t^{\text{dense}}, I, t). \tag{2}$$

After obtaining $x_0^{\text{dense}}$, the dense-stage VAE decoder maps it to an occupancy grid $\mathcal{O}^{\text{dense}} \in \mathbb{R}^{64 \times 64 \times 64}$.

**Stage 2: 512 Resolution** We select sparse voxel coordinates $\text{Index}^{\text{512-stage}} \in \mathbb{N}^{N_1 \times 3}$ at locations where $\mathcal{O}^{\text{dense}} > 0.2$, where $N_1$ is the number of selected voxels. We then sample the initial sparse noise as the feature of the sparse voxel:

$$x_t^{\text{512-stage}} \in \mathbb{R}^{N_1 \times 16} \sim \mathcal{N}(0, 1). \tag{3}$$

The sparse latent is denoised iteratively as

$$x_{t-1}^{\text{512-stage}} = x_t^{\text{512-stage}} - \epsilon^{\text{512-stage}}(\{x_t^{\text{512-stage}}, \text{Index}^{\text{512-stage}}\}, I, t). \tag{4}$$

Next, $x_0^{\text{512-stage}}$ is decoded into an SDF grid $\mathbf{S_{512}} \in \mathbb{R}^{512 \times 512 \times 512}$. Then, the 512-stage mesh is extracted from $\mathbf{S_{512}}$ using the marching cube algorithm.

**Stage 3: 1024 Resolution** We form $\text{Index}^{\text{1024-stage}} \in \mathbb{N}^{N_2 \times 3}$ by selecting grid points whose distance to the 512-stage surface is below a threshold, where $N_2$ is the number of selected voxels. We sample:

$$x_t^{\text{1024-stage}} \in \mathbb{R}^{N_2 \times 16} \sim \mathcal{N}(0, 1). \tag{5}$$

Then, the sparse latent is denoised iteratively by

$$x_{t-1}^{\text{1024-stage}} = x_t^{\text{1024-stage}} - \epsilon^{\text{1024-stage}}(\{x_t^{\text{1024-stage}}, \text{Index}^{\text{1024-stage}}\}, I, t). \tag{6}$$

Next, $x_0^{\text{1024-stage}}$ is decoded into an SDF grid $\mathbf{S_{1024}} \in \mathbb{R}^{1024 \times 1024 \times 1024}$. Finally, the final mesh is extracted from $\mathbf{S_{1024}}$ using the marching cube algorithm.

### 3.1.3 3D RECONSTRUCTION MODELS

We now introduce the point-map-based 3D reconstruction model. We choose the currently most robust model, VGGT (Wang et al., 2025a), as the base model. Given the input image $I$, VGGT model can predict a 3D coordinate of point's position for each pixel in the image with a confidence.

$$\mathcal{P}_{init}, \mathcal{C} = \text{VGGT}(I) \tag{7}$$

Then, we filter out the low-confidence area by a threshold to get the single-layer object shape from the input direction.

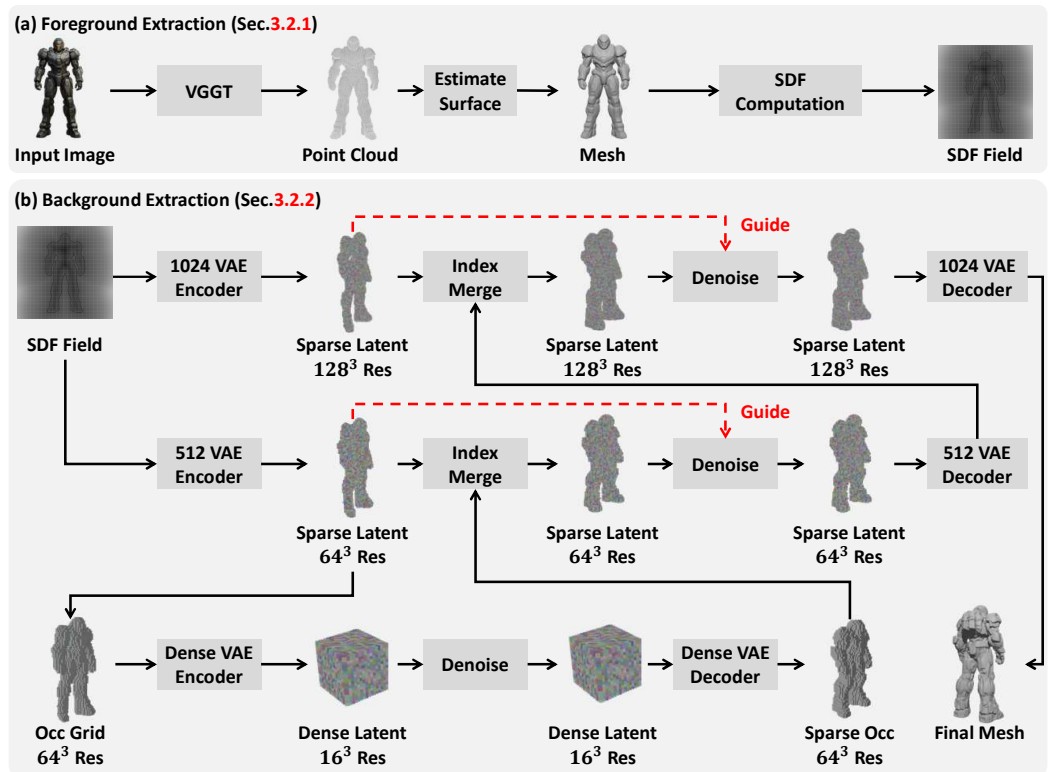

Figure 2: The framework of Consistent3DGen. (a) We first use the point map prediction model, VGGT, to get the foreground target and transform it into an SDF Field. (b) Next, we encode it into the latent space and use this information to guide the back completion.

## 3.2 CONSISTENT3DGEN

We argue that the image-to-3D task is equal to the combination of reconstruction at known views and generation at unknown views. Next, we introduce how we implement this training-free framework based on this idea.

### 3.2.1 FOREGROUND EXTRACTION

As illustrated in Fig. 2 (a), our inference pipeline proceeds as follows. Given an input image $I$, we first apply VGGT to predict a dense point map $\mathcal{P}$. To obtain a continuous surface representation amenable to downstream processing, we convert the point map, an $[H, W, 3]$ dimension tensor, into a triangle mesh. Specifically, we connect neighboring pixels into faces wherever a validity mask is satisfied. The mask is obtained from the confidence in the VGGT prediction and the sudden change of coordinates. The face-construction routine is detailed in Alg. 1. Finally, we can compute the SDF value using the estimated mesh surface.

### 3.2.2 BACK COMPLETION

As shown in Fig. 2 (b), this section explains how we inject reconstruction signals into the diffusion denoising process. We start from the SDF Field obtained from the last section. Firstly, we use the 512-stage VAE encoder and the 1024-stage VAE encoder to encode the SDF field and cut off the back-half part to get the target sparse latent $x_{fg}^{512\text{-stage}}$ and $x_{fg}^{1024\text{-stage}}$ for the direction of the user input, whose spatial resolution are $64^3$ and $128^3$ respectively. Next, we use the dense-stage VAE encoder to encode the occupancy grid, which is from the sparse index of $x_{fg}^{512\text{-stage}}$, into dense latent $x_{fg}^{\text{dense}}$, whose spatial resolution is 16. We then use this latent as the initial latent to call the dense-stage DiT to denoise this latent for completing the missing part of this object. After decoding the

---

**Algorithm 1** Mesh Face Generation Algorithm

---

**Require:** $rows$: number of grid rows, $cols$: number of grid columns, $mask$: boolean mask array
**Ensure:** $faces$: list of triangular face indices
 1: $faces \leftarrow \emptyset$                                       ▷ Initialize face list
 2: **for** $i \leftarrow 0$ **to** $rows - 2$ **do**
 3:     **for** $j \leftarrow 0$ **to** $cols - 2$ **do**
 4:         $p_1 \leftarrow i \times cols + j$                         ▷ Top-left vertex
 5:         $p_2 \leftarrow i \times cols + (j + 1)$               ▷ Top-right vertex
 6:         $p_3 \leftarrow (i + 1) \times cols + j$              ▷ Bottom-left vertex
 7:         $p_4 \leftarrow (i + 1) \times cols + (j + 1)$      ▷ Bottom-right vertex
 8:         **if** $mask[i][j] \wedge mask[i][j + 1] \wedge mask[i + 1][j] \wedge mask[i + 1][j + 1]$ **then**
 9:             $faces.\text{APPEND}([p_1, p_2, p_3])$            ▷ First triangle
10:             $faces.\text{APPEND}([p_2, p_3, p_4])$         ▷ Second triangle
11:         **end if**
12:     **end for**
13: **end for**
14: **return** $faces$

---

denoised latent, we can obtain a new occupancy grid with $64^3$ spatial resolution. We compute the union set of the index where occupancy is greater than $0.5$ and the index of $x_{fg}^{512\text{-stage}}$ as the sparse index $\text{Index}^{512\text{-stage}} \in \mathbb{R}^{N_1 \times 3}$. According to these sparse positions, we randomly sample the initial latent use Eq. 3. But different from the original Direct3D-S2 (Wu et al., 2025), we design a method to partially guide the denoising process using the previously encoded surface latent $x_{fg}^{512\text{-stage}} \in \mathbb{R}^{N_1' \times 16}$, where $N_1' < N_1$. Specifically, we denote the position mask $\mathcal{M} \in \mathbb{R}^{N_1}$, where the position in $x_{fg}^{512\text{-stage}}$ is set to 1 and the other is set to 0. In this stage, the original flow-matching denoising equation could be written as Eq. 4. Since we have the part target, we could obtain the partial update direction at timestep $t$ as

$$\Delta_t = (x_t^{512\text{-stage}} \odot \mathcal{M}) - x_{fg}^{512\text{-stage}}. \tag{8}$$

Therefore, to iteratively influence the final result, *i.e.*, making the partial area generate our wanted shape with the other area is denoised naturally, our final denoising equation is as follows

$$x_{t-1}^{512\text{-stage}} = x_t^{512\text{-stage}} - (\epsilon^{512\text{-stage}}(\{x_t^{512\text{-stage}}, \text{Index}^{512\text{-stage}}\}, I, t) + \lambda \Delta_t), \tag{9}$$

where $\lambda \in \mathbb{R}$ is a coefficient to control the guiding strength. For 1024-stage denoising, the procedures are similar to those of the 512-stage. After these injections, we can get a consistent object at the input direction.

### 3.2.3 DISCUSSION

**How do we encode a single-layer mesh face into SDF latent space in Direct3D-S2?**

Since we can only obtain a single-layer surface from Algorithm 1, it does not meet the requirements of a watertight mesh for Direct3D-S2's VAE. To address this, we first calculate the unsigned distance. Then, we select the area where the distance is less than a threshold, *i.e.*, the sparse voxels near the surface. We only use these sparse voxels to finish the later computation. As illustrated in Fig. 3, within the narrow area near the surface, we then normalize these values to the range of $[-1, 1]$, where the zero-level set emerges naturally. This transformation converts the simple surface into a thin, watertight mesh. As shown in Fig. 4, the VAE trained by Direct3D-S2 can encode this mesh successfully.

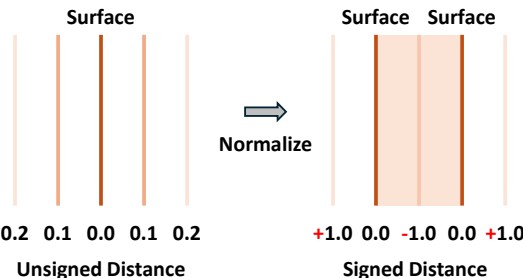

Figure 3: We transform the single-layer surface to a thin and watertight mesh by normalizing the unsigned distance value to $[-1, 1]$.

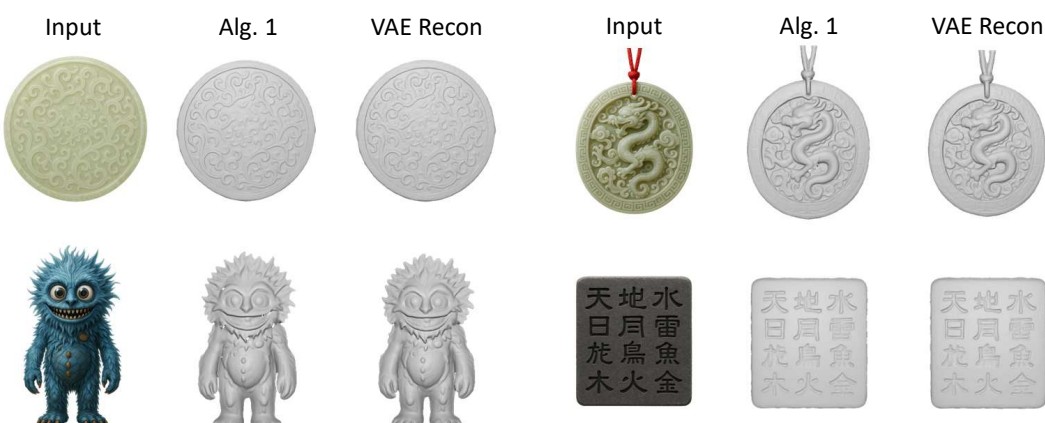

Figure 4: Evaluation of VAE Reconstruction.

**Why do we use the front-half part of the sparse latent?**

As discussed in the previous paragraph, we convert the single-layer surface into a thick, watertight mesh. However, our goal is to retain only the front surface of this mesh. Since the latent space relies on an explicit sparse voxel representation, we can directly use the front half of the mesh as the target partial latent. Additionally, because the SDF has a certain thickness, failing to cut away the rear half would result in a noticeable gap between the reconstructed back surface and the front surface.

## 4 EXPERIMENTS

### 4.1 IMPLEMENTATION DETAILS

All experiments are conducted on a single NVIDIA A40 48GB GPU. For the foreground surface point cloud extraction, we utilize the official VGGT-1B model, which processes $518 \times 518$ images as input. The back completion is performed using Direct3D-S2 V1.1. The foreground Signed Distance Field (SDF) extraction takes approximately 10 seconds. For the back completion, we apply the same strategy as in Direct3D-S2, reducing the number of mesh faces using a remeshing algorithm with a simplification coefficient of 0.95. As a result, the generated meshes typically contain between 300,000 and 1,000,000 faces. Consequently, the back completion process takes about 3-5 minutes, which is consistent with the time required by Direct3D-S2.

### 4.2 EVALUATION OF VAE RECONSTRUCTION

As shown in Fig. 4, to demonstrate that our algorithm could transform the point-map prediction into the 3D latent space of Direct3D-S2, we show the input image, VGGT processed output, and the reconstructed mesh that was first encoded by the VAE encoder, and then decoded by the VAE decoder. Our algorithm has proven effective in generating a reliable face-forward latent representation, with the reconstructed mesh closely matching the input image in a nearly one-to-one manner. The reconstructed mesh match the input image nearly in a one-to-one manner. Additionally, for some unstable outputs from VGGT, such as test case 3 (the monster) in Fig.4, small holes appear in the point map. Surprisingly, they are removed in the VAE reconstructed mesh, which demonstrates a strong robustness. Remarkably, these holes are eliminated in the VAE-reconstructed mesh, demonstrating the algorithm's robustness. These results lay a strong foundation for the subsequent guided denoising process.

### 4.3 OVERALL COMPARISONS

As shown in Fig. 5, we provide a comparison with recent open-sourced SoTA 3D generative models, including Trellis (Xiang et al., 2025), Hi3DGen (Ye et al., 2025), and Direct3D-S2 (Wu et al., 2025). A key observation is that direct generation often leads to severe hallucinations. For instance, in

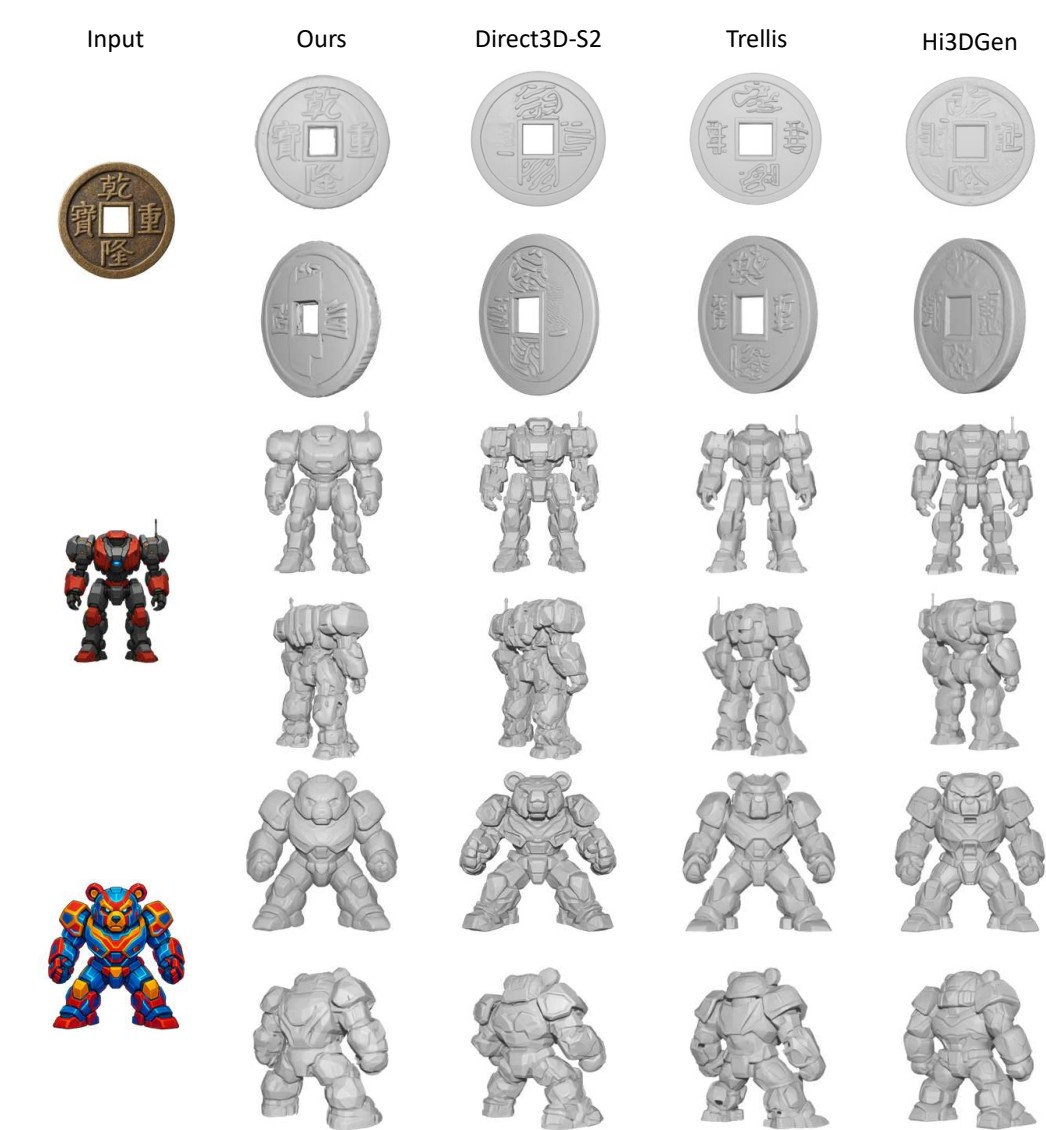

Figure 5: Overall comparisons with Direct3D-S2 (Wu et al., 2025), Trellis (Xiang et al., 2025), and Hi3DGen (Ye et al., 2025).

the first test case, where an ancient coin with four Chinese characters is used as input, all three baseline methods fail to generate the correct characters. While they place symbols in the correct positions, they do not match the intended characters. In contrast, our approach ensures pixel-aligned foreground generation, allowing for a precise one-to-one correspondence between input images and generated objects. In the second test case, other methods not only fail to map the geometric details accurately from a frontal perspective but also introduce extraneous features that are not present in the input. Our method, on the other hand, accurately preserves the geometric consistency on the front side and generate the corresponding structures on the back side. In addition to the aforementioned methods, we also include comparisons with closed-source approaches in the supplementary material.

### 4.4 ABLATION STUDIES

**The effect of cutting off half of the sparse voxel.** As shown in Fig. 6, we validate the conclusion of the discussion in Sec. 3.2.3. If we directly use the whole foreground latent, there will be an obvious

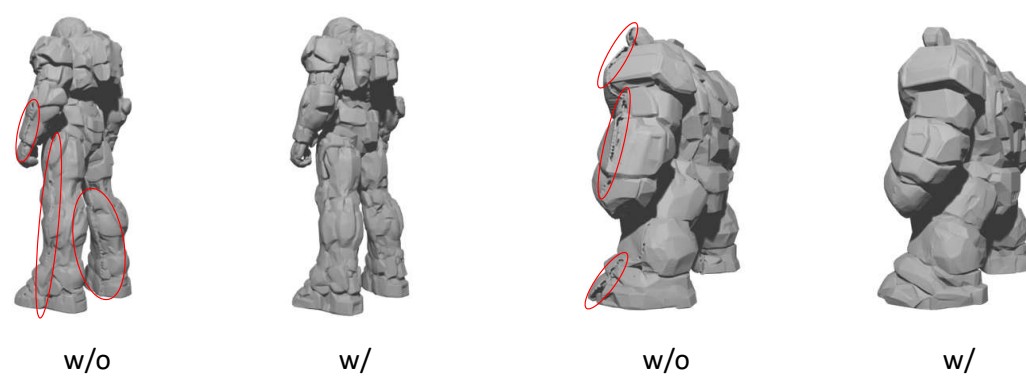

Figure 6: Ablation studies for the strategy of cutting half the foreground voxels. "w/o" represents "without the cutting-off strategy". "w/" represents "with the cutting-off strategy".

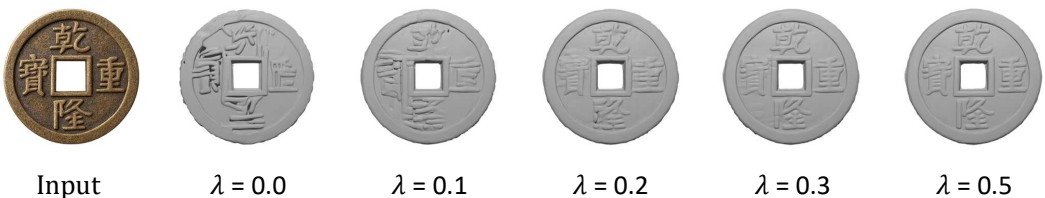

Figure 7: Ablation studies for the foreground guiding strength $\lambda$.

gap between the foreground and the background. To maintain the naturalness of the connection, we choose to cut off the back half of the voxel.

**The effect of guiding strength** $\lambda$. As shown in Fig. 7, we demonstrate the face-forward surface denoising result. As the guiding strength $\lambda$ increases, we can observe the text gradually becoming more and more consistent with the input image. To minimize the impact on the entire system, we select values that no longer change during convergence.

## 5 CONCLUSION

Current 3D generative models often struggle to generate detail-aligned and consistent 3D geometry in correspondence with the input image. To address this issue, we propose a training-free approach that mitigates these challenges. We assert that the image-to-3D task could be viewed as foreground reconstruction with background completion. To achieve this, we leverage the recent 3R model, which generates a pixel-aligned point map, *i.e.*, point cloud corresponding to the input image. Building on this idea, we design an algorithm that maps the foreground point cloud into the latent space of the 3D VAE. Subsequently, we employ a paired 3D diffusion model to complete the geometry of the object's back side. Theoretically, our framework is adaptable to any point map prediction model and sparse-voxel-based 3D diffusion model. As both types of models continue to advance, the performance of our framework is expected to improve accordingly.

**Limitations.** As a training-free framework, our method relies heavily on the performance of the base models. For instance, the current base model, VGGT, faces challenges with high-frequency reconstruction, due to the limitation of $518 \times 518$ input resolution. Therefore, our method requires better base models for further improvements.

**Future Work.** Besides developing the training-free methods, we can improve the overall performance by 1) finetuning VGGT for better object-level performance, and 2) developing a specialized completion network to accomplish the back completion task.

## Reproducibility Statement

Upon acceptance of the paper, we will release the project code. The code will be made available in a public repository, such as GitHub, to ensure transparency and accessibility for future researchers. Additionally, we will provide clear documentation, including setup instructions, dependencies, and detailed explanations of the key algorithms and models used in the study. This will allow others to easily replicate our experiments, validate our results, and build upon the work presented in this paper. The results of all other methods are from their official repository or website.

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

# A    COMPARISONS WITH CLOSED-SOURCED METHODS

We also compare the generation quality with recent state-of-the-art closed-source 3D generative models, including Sparc3D (Li et al., 2025b), Hunyuan3D-2.5 (Lai et al., 2025), and Ultra3D (Chen et al., 2025). As shown in Fig. 8, these models exhibit impressive detail and significantly higher consistency compared to recent open-source methods. However, they still fail to generate accurate Chinese characters. While characters with simple glyphs are generated correctly, inconsistencies persist. In contrast, more complex characters are often generated incorrectly (as indicated by the red circle in Fig. 8). In comparison, our method, which utilizes pixel-aligned foreground surfaces, achieves much higher consistency.

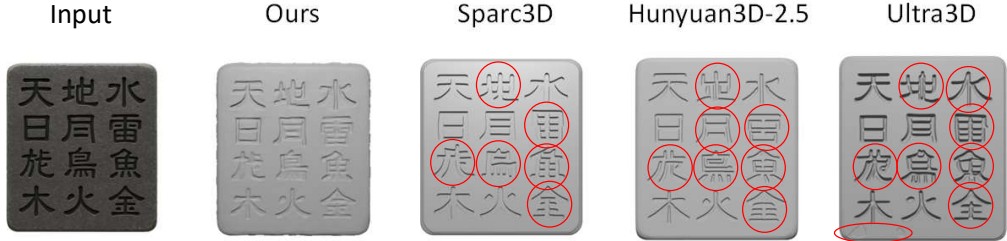

Figure 8: Comparison with closed-sourced methods.

# B    THE USE OF LARGE LANGUAGE MODELS (LLM)

We used OpenAI's GPT-5 to assist with the refinement and proofreading of certain sentences in this paper. The LLM was used exclusively to enhance the clarity and coherence of our writing. All content contributions are made by the authors.

