# OpenReview forum: "Consistent3DGen: Bridging Stochastic Generation and Deterministic Reconstruction for Image-to-3D Diffusion Models"
_ICLR.cc/2026/Conference — ICLR 2026 Conference Withdrawn Submission_

### Official Review · Reviewer_x5qt · 2025-10-28

**Soundness:** 2
**Presentation:** 2
**Contribution:** 2
**Rating:** 2
**Confidence:** 5

**Summary:**

Although 3D diffusion methods yield remarkable details, they frequently deviate from the precise alignment of the input images. In contrast, feedforward methods typically achieve a higher level of alignment with the input image. However, the back surfaces are often excessively smoothed and lack the necessary detail. The authors propose a training-free method to establish this optimal alignment for the frontal surface using a point-cloud prediction network (VGGT) coupled with a diffusion-based back surface generation process. For the back surface direction, the authors suggest employing Direct3D-S2. To incorporate the frontal predictions, the authors initially estimate the surface from the VGGT point clouds. Subsequently, they compute an SDF, which serves as the initial point for the denoising process of Direct3D-S2.

**Strengths:**

- The integration of feed-forward and diffusion techniques in a training-free approach is highly interesting.

**Weaknesses:**

- The presented method lacks thorough evaluation. It fails to provide metrics or quantitative comparisons with existing works. This make judging it accurately impossible.
- The results presented in Fig. 5 are questionable. The frontal view is significantly less detailed than the back view. Although the alignment is stronger, the disparity in quality levels is significant, resulting in a diminished overall asset quality.
- An alternative method, SPAR3D, proposed by Huang et al., combines feed-forward and diffusion techniques with a similar reasoning. SPAR3D begins with a point cloud diffusion and utilizes it in a feed-forward mesh estimator. The different approaches should be discussed in the related work section.
- Without prior knowledge of Direct3D-S2, the paper remains challenging to comprehend.

**Questions:**

Without a proper evaluation, it is hard to evaluate a paper - especially in an active field like 3D asset generation. Given this point alone, I can only recommend a rejection.

---

### Official Review · Reviewer_cDkw · 2025-10-30

**Soundness:** 2
**Presentation:** 3
**Contribution:** 2
**Rating:** 4
**Confidence:** 4

**Summary:**

This paper aims to address the issue that image-to-3D diffusion models often fail to maintain precise consistency with the input image. The authors identify the cause as a conflict between random generation and faithful reconstruction. To solve this, they propose Consistent3DGen, a training-free framework that separates these two tasks. First, a deterministic reconstruction model (VGGT) extracts a pixel-aligned front-view point cloud. This point cloud is then converted into a partial VAE latent representation (Sec. 3.2.3). Finally, a new front-partial denoising guidance mechanism (Eq. 9) directs the base diffusion model (e.g., Direct3D-S2) to follow the front latent representation while generating the unseen back view. Experiments show that this method preserves foreground details with high consistency.

**Strengths:**

- The paper presents a clear and effective approach by decoupling Image-to-3D generation into deterministic front reconstruction and generative back completion. It is easy to implement and requires no extra training, making it highly practical.
- The proposed method effectively mitigates the inconsistency between the generated 3D content and the original input image, a common issue in single-image-to-3D generation pipelines. The visual results demonstrate significant improvements in preserving details and maintaining fidelity to the input image.

**Weaknesses:**

- The paper relies entirely on qualitative visual comparisons to support its claims of superior consistency (Sec. 4.3, Fig. 5, Fig. 8). It fails to report any quantitative metrics (e.g., PSNR, LPIPS, Chamfer distance, or other standard 2D/3D evaluation metrics) to measure the rendering quality and consistency of the generated 3D models against the input images. This lack of quantitative evaluation makes it difficult to objectively assess the effectiveness of the proposed method compared to existing approaches.

**Questions:**

- The shown cases are all rendered images. Could you provide comparisons using real photographs to better demonstrate the real-world applicability of your method?

- Although the author explains that VGGT was used because of its reconstruction certainty, the input viewpoint is a single view. In this case, the generated point cloud is basically the same as monocular depth estimation. However, VGGT does not have a significant advantage in monocular depth estimation. So why not directly use other state-of-the-art monocular depth estimation models such as Depth Anything [1] to generate the point cloud? This could also address the limitation on input resolution.

[1] Depth Anything V2. A More Capable Foundation Model for Monocular Depth Estimation

---

### Official Review · Reviewer_dQWc · 2025-10-30

**Soundness:** 1
**Presentation:** 2
**Contribution:** 1
**Rating:** 2
**Confidence:** 5

**Summary:**

This paper proposes to use the point-cloud predicted by VGGT for a single input frontal canonical view (of 3D object) to obtain the front part of the 3D mesh, and then inpaint the occluded back part with Direct3D-S2.

Unfortunately, the proposed method lacks some crucial details in the presentation and presents no quantitative evaluation compared to existing regressive/generative 3D reconstruction approaches such as Hunyuan3D [1], LGM [2], InstantMesh [3] or LucidFusion [4]. (More details in weaknesses)

Thus other than the obvious limitation of requiring canonical frontal views, it's impossible to asses the effectiveness and the utility of the proposed approach.

[1] Zhao, Z., Lai, Z., Lin, Q., Zhao, Y., Liu, H., Yang, S., ... & Guo, C. (2025). Hunyuan3d 2.0: Scaling diffusion models for high resolution textured 3d assets generation. arXiv preprint arXiv:2501.12202.

[2] Tang, J., Chen, Z., Chen, X., Wang, T., Zeng, G., & Liu, Z. (2024, September). Lgm: Large multi-view gaussian model for high-resolution 3d content creation. In European Conference on Computer Vision (pp. 1-18). Cham: Springer Nature Switzerland.

[3] Xu, J., Cheng, W., Gao, Y., Wang, X., Gao, S., & Shan, Y. (2024). Instantmesh: Efficient 3d mesh generation from a single image with sparse-view large reconstruction models. arXiv preprint arXiv:2404.07191.

[4] He, H., Liang, Y., Wang, L., Cai, Y., Xu, X., Guo, H. X., ... & Chen, Y. (2024). LucidFusion: Reconstructing 3D Gaussians with Arbitrary Unposed Images. arXiv preprint arXiv:2410.15636.

**Strengths:**

1. The paper highlights an important and relevant problem of being able to control the 3D generative process of foundational models like Direct3D-S2 or Trellis.

2. The introduction and related work has good flow and coverage.

**Weaknesses:**

**Major concerns**:
1. No quantitative experimental evaluation is presented, which makes it impossible to assess the utility of the proposed approach. Please refer to the standard reconstruction evaluation setups (scale-normalization and sampling ratios) with metrics such as Chamfer-Distance or F-score from various 3D reconstruction or NVS setups.

2. An important detail about how exactly the denoising at the dense stage is done is missing? Do you only denoise from a smaller timestep or if the same frontal CFG, similar to the 512 and 1024 stages, is enforced?

3. Lastly, the biggest limitation off the proposed method is that a frontal canonical view is needed for 3D reconstruction which is not always available, or another generative model would need to be used to map arbitrary images to frontal canonical, which can again cause generative drifting.

**Minor concerns/mistakes**
1. Equation 2, scaling by $(t_k - t_{k-1})$ is missing in the Euler sampling process. Same with equation 4.

**Questions:**

Unfortunately, the paper lacks major crucial details in the presentation and presents no quantitative evaluation compared to existing regressive/generative 3D reconstruction approaches, because of which it is not possible to assess where the 3D reconstruction line of research this proposed method fits.

Thus, a reject is recommended since the prevailing issues cannot be fixed within the scope of this submission cycle.

---

### Official Review · Reviewer_VRpK · 2025-10-31

**Soundness:** 3
**Presentation:** 2
**Contribution:** 3
**Rating:** 6
**Confidence:** 3

**Summary:**

This paper introduces Consistent3DGen, a training-free framework designed to improve the view-consistency of image-to-3D diffusion models. Current 3D diffusion models (e.g., Direct3D-S2, Trellis, Hi3DGen) can produce high-fidelity 3D assets but often fail to maintain geometric consistency with the input image due to stochastic initialization noise .

The key insight is to decouple deterministic reconstruction (visible regions) from stochastic generation (occluded regions). The method first uses a pixel-aligned 3D reconstruction model (VGGT) to obtain an accurate front-facing point cloud. This is then mapped into the latent space of a pre-trained 3D diffusion model (Direct3D-S2) via a novel point-to-latent mapping algorithm. During denoising, a front-partial guidance mechanism constrains visible-region latents while letting the diffusion model freely complete the unseen back geometry .

**Strengths:**

- Clear motivation: Articulates the tension between stochastic generation and faithful reconstruction and addresses it conceptually and technically.

- Elegant training-free formulation: Does not require finetuning or retraining large 3D diffusion models - only adds deterministic geometric constraints.

- Novel front-partial denoising guidance: A simple yet effective idea to fuse deterministic priors with generative flexibility.

- Strong qualitative improvements: Results in Figs. 5 & 8 show clear alignment between input and generated geometry, with markedly fewer hallucinations (e.g., legible Chinese characters, correct shape correspondences).

**Weaknesses:**

- Limited quantitative evaluation: The comparisons are mainly visual; no numerical metrics (e.g., FID-3D, PSNR, consistency scores) are reported.

- Dependency on strong base models: Performance is bounded by VGGT’s front-surface accuracy and Direct3D-S2’s latent expressivity.

- No discussion of failure cases: E.g., what happens if VGGT reconstruction is imperfect or contains holes.

= Lack of scalability benchmarks: The method’s behavior on complex scenes or multi-object compositions is not analyzed.

**Questions:**

- Runtime is reported as “3–5 minutes per mesh” - could the authors break down the timing between VGGT inference, latent encoding, and diffusion stages for transparency?

- How does Consistent3DGen perform on multi-object or cluttered scenes versus single-object inputs?

- Would training a lightweight alignment module (e.g., learned latent mapper or score distillation fine-tune) further improve consistency, or does it risk losing the “training-free” simplicity?

---

### Note · Authors · 2025-11-13

**Comment:**

We sincerely thank all the reviewers for their contributions at this stage. We will revise our paper.

**Withdrawal Confirmation:**

I have read and agree with the venue's withdrawal policy on behalf of myself and my co-authors.